# Inheriting Generalizable Knowledge from LLMs to Diverse Vertical Tasks

**Chang Liu[1,2], Boyu Shi[1,2], Xu Yang[1,2],** * **Qiufeng Wang[1,2], Xin Geng[1,2]** †

[1]School of Computer Science and Engineering, Southeast University, Nanjing, China
[2]Key Laboratory of New Generation Artificial Intelligence Technology and
Its Interdisciplinary Applications (Southeast University), Ministry of Education, China
{liuchang0520,shiboyu,xuyang_palm,qfwang,xgeng}@seu.edu.cn

## Abstract

Large language models (LLMs) have demonstrated remarkable generalization across diverse tasks, suggesting the existence of task-agnostic, generalizable knowledge encoded within them. However, how to systematically extract and evaluate this knowledge remains unexplored. In this work, we innovatively propose MASA (**M**atrix-level **A**lignment and **S**calable **A**daptation), a unified framework for extracting and transferring generalizable knowledge from LLMs. MASA first introduces a lightweight set of square matrices trained with a dual alignment strategy, combining output alignment and spectral alignment, to capture the generalizable knowledge encoded in the Feed-Forward Network (FFN) of LLM. It then employs scalable adaptation to flexibly reshape these matrices to match the parameter dimensions of lightweight dense models of various sizes, enabling direct initialization of their FFN layers. To evaluate the inherited knowledge, we measure the downstream performance of lightweight models initialized with MASA across language understanding and dialogue generation tasks spanning diverse vertical domains. Experiments on both dense and Mixture-of-Experts (MoE) source LLMs show that MASA consistently outperforms baselines such as random initialization, distillation and pruning, yielding lightweight models that achieve stronger performance, require less pre-training data, and converge faster. These results establish MASA as an effective and general framework for extracting and leveraging the generalizable knowledge within LLMs.

## 1 Introduction

Large language models (LLMs) (Wiggins & Tejani, 2022; Chowdhery et al., 2023; Achiam et al., 2023; Grattafiori et al., 2024; Wang et al., 2025; Zhong et al., 2025) have demonstrated remarkable generalization across a wide range of tasks, due to their ability to learn rich representations. Several lines of evidence suggest that this capability stems from the presence of meta-level and task-agnostic knowledge within the models. Parameter-efficient fine-tuning methods such as LoRA (Hu et al., 2022) and adapters (Houlsby et al., 2019) show that by only adjusting a small subset of parameters (or adding lightweight modules), large models can be adapted to new tasks with minimal cost. This implies that the core of task-agnostic and generalizable knowledge is already embedded in the fixed pre-trained components of the model. Similarly, instruction tuning (Wei et al., 2021) and in-context learning (Brown et al., 2020) indicate that LLMs possess a reservoir of general capabilities that can be flexibly applied across domains and tasks, further supporting the hypothesis that LLMs encode a condensed core of meta-level and generalizable knowledge.

Taken together, these evidence highlight the existence of generalizable and task-agnostic knowledge within LLMs, motivating investigations into its structural localization. Previous studies have shown that the Feed-Forward Network (FFN) layers of transformers encode richer and more generalizable knowledge than other components (Geva et al., 2021; Ikeda et al., 2025), a finding further reinforced by the success of Mixture-of-Experts (MoE) architecture (Team et al., 2024; Guo et al., 2025;

---

*Corresponding Author.
†Corresponding Author.

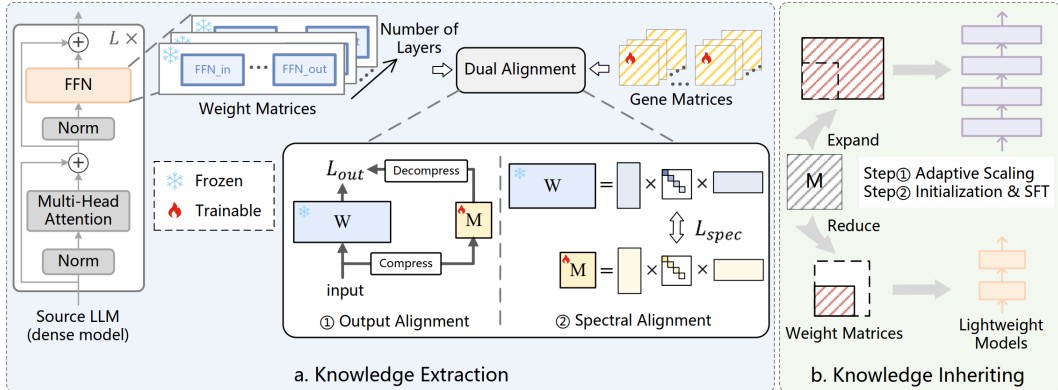

Figure 1: Overview of the MASA framework. Generalizable knowledge is first captured in gene matrices through alignment with the source LLM, and then these matrices are adaptively scaled to initialize lightweight models. In this figure, we take a dense-architecture source LLM as an example.

Muennighoff et al., 2025), which is typically applied in FFN layers. In MoE models, experts that are frequently activated across diverse tasks are often regarded as carriers of shared, task-agnostic knowledge (Dai et al., 2024; Li et al., 2025c), providing additional evidence that FFNs naturally embody generalizable representations.

Although previous studies have established the existence of generalizable knowledge in LLMs and identified FFN layers as its primary locus, **little work has explored how to explicitly extract and leverage such knowledge**. In the vision domain, a representative attempt in this direction is the *Learngene* (Wang et al., 2023a) framework. Learngene extracts common knowledge across tasks from large Vision Transformers (ViTs) and conceptualizes this knowledge module as learngene, which is believed to embody meta-level and generalizable capabilities. When the knowledge module is used to initialize smaller models, it enables smaller architectures to achieve stronger generalization and superior performance on downstream tasks, thereby unlocking greater potential than conventional initialization methods. A learngene may correspond to an entire block (Wang et al., 2023a) or a parameter submatrix within a block (Xia et al., 2024), capturing transferable knowledge distilled from the well-pretrained large model. However, the Learngene framework has only been applied to small ViTs in the vision domain, leaving the extraction of meta-level, generalizable knowledge from LLMs largely unexplored.

Inspired by the Learngene framework for extracting generalizable and task-agnostic knowledge from models, we aim to extract and reuse such knowledge from LLMs for the first time. In this work, we innovatively propose **MASA** (**M**atrix-level **A**lignment and **S**calable **A**daptation), a unified framework for extracting and transferring generalizable knowledge from LLMs.

Specifically, as illustrated in Figure 1 a, we train a set of square matrices, termed gene matrices, to mimic the FFN weight matrices in each block of the LLM. Through this process, the gene matrices effectively capture the generalizable knowledge of the model. During training, only the gene matrices are updated, while the parameters of the LLM remain fixed. To ensure faithful knowledge extraction, we introduce output alignment, which enforces consistency between the outputs of the gene matrix and the corresponding LLM weight matrix. In addition, motivated by the observation that the singular value distribution of parameter matrix correlates with the generalization performance of model (Mahoney & Martin, 2019; Yoshida & Miyato, 2017; Martin et al., 2021), we propose spectral alignment to capture structural characteristics associated with generalizable knowledge. Moreover, to further enrich the extracted knowledge and enhance its generalization, we train the gene matrices on data spanning a wide range of domains, including but not limited to GitHub, arXiv, and Wikipedia.

Since no previous work has proposed a method for leveraging the generalizable knowledge in LLMs, we propose a reusing framework that transfers the extracted knowledge into lightweight dense models as initialization. To enable flexible transfer across models of arbitrary dimensions, we further design an **adaptive scaling** strategy, which reshapes the gene matrices while preserving their representational structure, thus making them suitable for initializing lightweight models with diverse

sizes. The downstream performance of these models then directly reflects the quality of the extracted knowledge. As shown in Figure 1 b, this method flexibly expands or compresses fixed-size gene matrices to match the parameter dimensions of target dense models, enabling direct use for initialization. Specifically, each matrix is decomposed via singular value decomposition, retaining its diagonal eigenvalue matrix. The singular vectors are then trimmed or padded along rows and columns to reshape the matrix while preserving its structural representation and knowledge capacity. The adapted matrices are then aligned with the parameter dimensions of the target dense model and directly used to initialize its FFN layers, while the remaining parameters are randomly initialized.

Just as prior work has not explored how to extract and reuse the generalizable knowledge of LLMs, no evaluation protocol has been established for assessing such knowledge. We therefore propose a novel evaluation protocol: an effective method should enable lightweight models which inherit knowledge from LLM to **achieve stronger performance across diverse tasks**, **require less pre-training data**, and **converge more rapidly**. We compare against representative baselines including random initialization, knowledge distillation, and pruning.

To comprehensively evaluate the extracted knowledge, we conduct experiments using three representative source LLMs: dense OLMo-7B (Groeneveld et al., 2024), standard MoE OLMoE-7B (Muennighoff et al., 2025), and shared-expert MoE DeepSeekMoE-16B (Dai et al., 2024), covering all major LLM architectures. For each of these source LLMs, the total parameters of all gene matrices used for knowledge extraction range only **11.8M-38.6M** parameters, accounting for as little as **0.17%** of the source LLMs. Thus, training the gene matrices requires a small amount of data—just **4M-10M** tokens are sufficient for convergence. To test the effectiveness of the extracted knowledge, we initialize lightweight models ranging from 267M to 877M parameters with gene matrices and evaluate them on diverse vertical-domain supervised fine-tuned (SFT) datasets. Remarkably, these models can achieve over **85%** of the source LLM's performance after only **2B–10B** tokens of pre-training. Compared with baselines such as random initialization, pruning, and knowledge distillation, models that inherit LLM knowledge through the MASA approach consistently achieve stronger performance, lower data requirements, and faster convergence. These results demonstrate that MASA can effectively extract and transfer generalizable knowledge from LLMs to smaller ones, enabling more efficient and scalable model training.

## 2 RELATED WORK

**Generalizable Knowledge of LLMs.** Recent research increasingly investigates whether large language models (LLMs) (Li et al., 2025a;b; Huang et al., 2025; Peng et al., 2025; Li et al., 2024; Lu et al., 2025) encode generalizable knowledge that can be transferred across tasks (Wu et al., 2025). Scaling analyses and emergent ability studies demonstrate that LLMs acquire task-agnostic structures supporting zero-shot and few-shot generalization (Wei et al., 2022a; Srivastava et al., 2023). Beyond language modeling, such abilities extend to reasoning, planning, and domain adaptation (Achiam et al., 2023; Bai et al., 2022). To characterize generalizable knowledge, recent work employs probing, representation analysis, and intervention techniques, showing that transferable information often concentrates in modular components or subspaces (Geva et al., 2021; Meng et al., 2022; Allen-Zhu & Li, 2023). On the application side, parameter-efficient tuning methods, such as LoRA (Hu et al., 2022; Jiang et al., 2024) preserve generalization while reducing resource demands. Despite these advances, few works directly address how to systematically extract and reuse generalizable knowledge itself, which remains an open challenge and motivates our work.

**Knowledge Distillation and Pruning.** Knowledge distillation and pruning have been widely explored to reduce the size and computational cost of large language models. Distillation methods, such as MiniLLM (Gu et al., 2024) and EvoKD (Liu et al., 2024a), improve student models by better aligning with teacher outputs or leveraging task-specific active learning. Techniques such as in-context learning (Huang et al., 2022) and chain-of-thought distillation (Wei et al., 2022b; Wang et al., 2023b) further exploit LLMs' emergent capabilities for reasoning and downstream tasks. Pruning reduces model size by removing redundant components, either in a structured manner (Ma et al., 2023; Lee et al., 2024; Liu et al., 2024b)—removing experts, layers, or attention heads—or unstructured (Frantar & Alistarh, 2023; Xia et al., 2023)—eliminating individual weights. For sparse MoE models, specialized pruning approaches include efficient expert pruning (Liu et al., 2024b), retraining-free hierarchical merging (Chen et al., 2024), and task-agnostic expert diversifi-

cation (Zhang et al., 2024). Despite their effectiveness, both distillation and pruning face limitations when the scale gap between the source and target models is large: the transferred knowledge or pruned structure may fail to adapt, leading to structural degradation and reduced generalization.

**Learngene.** The Learngene framework was initially introduced by Auto-Learngene (Wang et al., 2023a). Auto-Learngene proposed a metalearning-based method that employed pseudo descendant models to identify learngene. By comparing each layer of a pseudo descendant with its counterpart in the source large model, their method evaluated layer similarity to determine which layers to retain. TLEG (Xia et al., 2024) further advanced the concept by revealing linear correlations between blocks in ViTs, enabling the construction of learngene through parameter-sharing linear combinations, thereby significantly reducing training overhead. Learngene Pool (Shi et al., 2024) developed a learngene pool using SN-net, which distills knowledge from the large source model into auxiliary models. These models were later composed into small models of various sizes via SN-net stitching.

## 3 METHOD

In this section, we detail the MASA framework. To extract generalizable knowledge from LLMs, MASA introduces a dual alignment mechanism: output alignment and spectral alignment. As shown in Figure 1 a, gene matrices are aligned with the source LLM's parameters in both function and structure, capturing generalizable and meta-level knowledge. In Figure 1 b, gene matrices are adaptively scaled to the lightweight model's parameter dimensions and used for initialization.

### 3.1 KNOWLEDGE EXTRACTION

In our method, we design a set of square matrices $G_M = \{M_{1,1}, \ldots, M_{l,k}, \ldots, M_{L,K}\}$ for the FFN layers in each block of the source LLM, where $L$ denotes the number of layers in the LLM and $K$ is the number of square matrices deployed per layer. Each matrix $M_{l,k}$ is responsible for tracking and aligning with one weight matrix of the LLM to extract its knowledge.

For standard MoE architecture such as OLMoE, we first measure the activation frequency of experts across different task datasets, and the gene matrices are aligned with those experts that are highly active across diverse tasks. For MoE structure with shared experts such as DeepSeekMoE, the gene matrices are aligned with the shared experts. In contrast, for dense models such as OLMo, the gene matrices are aligned with all parameter matrices in the FFN layers. In our experiments, the size of the matrix set ranges only **from 11.8M to 38.6M** parameters, with the smallest case accounting for merely **0.17%** of the source LLM's size.

To better extract the knowledge of LLMs, we propose a dual alignment mechanism for gene matrices. **Output Alignment** ensures that gene matrices can reproduce the functional behaviors of the source LLM's parameters, but this alone only captures surface-level mappings. To further inherit the intrinsic knowledge of LLMs, we employ **Spectral Alignment**, which focuses on the singular values of the parameter matrices. These singular values encode the essential structural patterns and generalization properties of the model (Mahoney & Martin, 2019; Yoshida & Miyato, 2017; Martin et al., 2021). By aligning the spectral structure, the matrices not only imitate the responses of LLM's parameters but also preserve their internal knowledge geometry, thereby providing the target lightweight model with a stronger foundation for generalization and downstream adaptation.

We denote by $W \in \mathbb{R}^{d_{in} \times d_{out}}$ any weight matrix within an LLM from which we aim to extract knowledge. Regardless of the architecture of the source LLM (whether dense or MoE) or the size of its parameter matrices, we can employ a compact matrix $M \in \mathbb{R}^{r \times r}$ to align with each weight matrix $W$. By applying the gene matrices $G_M$ in each layer, we are able to extract the knowledge embedded within the weight matrices of source LLM. However, since the input $x \in \mathbb{R}^{d_{in}}$ generally has a different dimensionality from $M$, the matrix cannot be applied directly to the input.

To address this, we first employ a compression function $f_c$ to reshape the input of dimension $d_{in}$ into a matrix of size $n \times r$, where $r$ corresponds to the rank of $M$ and $n = \lceil \frac{d_{in}}{r} \rceil$. After reshaping, the compressed input $f_c(x_{in})$ is multiplied by the matrix $M$. Finally, the resulting output is decompressed via $f_d$ to match the output dimension $d_{out}$ of the parameter matrix $W$. The entire process can

be formally expressed as a chain of operations (for more detailed procedures, see Appendix A.2):

$$\tilde{x} = f_d(M \cdot f_c(x_{in})) \in \mathbb{R}^{d_{out}}, \tag{1}$$

$$f_c(x) = [x_{1:r}, x_{(r+1):2r}, \ldots, x_{[(n-1)r+1]:nr}], \tag{2}$$

$$f_d(x) = \text{concat}(x). \tag{3}$$

For **output alignment**, we minimize the discrepancy between the responses of $W$ and $M$: $L_{out} = |Wx_{in} - \tilde{x}|^2$.

While output alignment captures the functional equivalence between two matrices, it only reflects surface-level mappings and ignores the internal knowledge structure embedded in the parameter matrices. To preserve this structural information, we introduce **spectral alignment**, which compares the singular values of the matrices. Since raw singular values often span several orders of magnitude, directly aligning them would be dominated by the largest values and overlook the relative decay pattern. To address this, we perform the alignment in the logarithmic domain, where the singular values form an approximately linear trend on a log-log scale. This emphasizes the decay shape rather than absolute magnitude, allowing the gene matrices to extract the LLM's intrinsic structural patterns that are crucial for generalization. Formally, given the singular value decomposition of weight matrix $W$ from LLM and matrix $M$ from $G_M$ as:

$$W = U\Sigma V^\top, \quad \Sigma = \text{diag}(\sigma_1, \sigma_2, \ldots, \sigma_r), \ \sigma_1 \geq \sigma_2 \geq \cdots \geq 0, \tag{4}$$

$$M = U'\Sigma'V'^\top, \quad \Sigma' = \text{diag}(\sigma_1', \sigma_2', \ldots, \sigma_r'), \ \sigma_1' \geq \sigma_2' \geq \cdots \geq 0. \tag{5}$$

The logarithmic spectral loss is defined as $L_{spec} = \sum_{i=1}^{r}(\log \sigma_i - \log \sigma_i')^2$. Here, the subtraction in the log domain ensures that the loss captures the relative spectral decay shape, rather than absolute differences in magnitude.

The final alignment objective combines the two as $L_{align} = L_{out} + \lambda L_{spec}$, where $\lambda$ controls the trade-off between functional equivalence and structural similarity.

**Theoretical Motivation: Spectrum and Generalization.** Recent studies indicate that the spectral properties of weight matrices, such as the distribution of singular values, are closely related to generalization ability. We provide a theoretical justification on why the spectral properties of model parameters are related to generalization (see details in Appendix A.1).

## 3.2 KNOWLEDGE INHERITING

To enable the reuse of generalizable knowledge extracted from LLMs, we propose to employ it for initializing lightweight models of varying scales. In the following, we describe how the extracted knowledge can be effectively utilized.

We introduce an **adaptive scaling** mechanism that allows the gene matrices to be flexibly adapted to target lightweight models of arbitrary dimensions. Given a certain matrix $M \in \mathbb{R}^{r \times r}$ from gene matrices, our goal is to transform it into a target matrix of arbitrary size $\hat{M} \in \mathbb{R}^{p \times q}$, where $p \times q$ denotes the weight matrix size in the FFN layer of the target lightweight model. The procedure consists of two stages: (1) decomposition into low-rank components and scoring the importance of them, and (2) row/column resampling by the scores to match the target size.

In the first stage, we perform the singular value decomposition on the matrix $M \approx U_r \Sigma V_r^\top$, where $U_r = [u_1^\top; u_2^\top; \ldots; u_r^\top] \in \mathbb{R}^{r \times r'}$, $\Sigma \in \mathbb{R}^{r' \times r'}$, $V_r = [v_1, v_2, \ldots, v_r] \in \mathbb{R}^{r' \times r}$, and $r'$ denotes the number of principal components retained. In this decomposition, the vector $u_i$ corresponds to the $i$-th row of $U_r$, and the vector $v_j$ corresponds to the $j$-th column of $V_r$. We then compute the norm of each row in $U_r$ and each column in $V_r$ as their respective importance scores:

$$s_i^{(U)} = ||u_i^\top||_2, i = 1, \ldots, r; \quad s_j^{(V)} = ||v_j||_2, j = 1, \ldots, r. \tag{6}$$

These scores reflect the contribution of the corresponding row or column to the subspace of the principal component.

Next, we describe the second stage, where we map the matrix $M \in \mathbb{R}^{r \times r}$ to a target matrix $\hat{M} \in \mathbb{R}^{p \times q}$. Taking the row-wise mapping as an example, we first compute the importance scores of each row from the singular value decomposition of $M$. These scores guide us in compressing or expanding the dimension of the row of $U_r$:

When $p \leq r$, we sort the row norms $s_i^{(U)}$ of $U_r$ in descending order and select the indices of the top $p$ rows, denoted as $\mathcal{I} = \{i_1, i_2, \ldots, i_p\}$. The compressed matrix is then constructed as $U_p = U_r[\mathcal{I}, :]$, where $U_r[\mathcal{I}, :]$ denotes the submatrix of $U_r$ formed by the rows indexed by $\mathcal{I}$.

When $p > r$, we sort the row norms $s_i^{(U)}$ of $U_r$ in descending order and select the indices of the top $(p - r)$ rows, denoted as $\mathcal{I} = \{i_1, i_2, \ldots, i_{p-r}\}$. These rows are then replicated and appended to $U_r$ until the target dimension is reached. Formally, we construct $U_p = \begin{bmatrix} U_r \\ U_r[\mathcal{I}, :] \end{bmatrix} \in \mathbb{R}^{p \times r'}$.

Similarly, the column-wise compression and expansion follow the same procedure as in the row-wise case, which allows us to map $V_r$ to the target matrix $V_q \in \mathbb{R}^{r' \times q}$. After obtaining $U_p$ and $V_q$, the target matrix $\hat{M} \in \mathbb{R}^{p \times q}$ is reconstructed as $\hat{M} = U_p \Sigma V_q^\top$. Through the above procedure, the matrix $M$ in the gene matrices can be adaptively rescaled into weight matrix of arbitrary size, which can then be directly used to initialize the FFN layers of target lightweight models with different scales. The remaining parameters of these lightweight dense models are randomly initialized. After a small amount of pre-training on limited data, the models are further fine-tuned on downstream datasets from different task types and vertical domains.

## 4 EXPERIMENTS

As outlined in the introduction, our goal is to extract generalizable knowledge from LLMs and reuse it in smaller ones, enabling rapid and efficient adaptation across sub-tasks. To this end, we compare MASA with multiple lightweight baselines and evaluate it across two sub-task dimensions: **task type**, including language understanding and dialogue generation, and **domain specialization**, covering various vertical domains and disciplines.

### 4.1 EXPERIMENTAL SETUP

**Data.** To ensure that the gene matrices in our framework adequately capture the generalizable knowledge from LLMs, we use **RedPajama-V2** Weber et al. (2024) as the training corpus for the gene matrices. RedPajama is an open and general-purpose LLM pre-training dataset, covering a wide range of knowledge domains, including but not limited to Wikipedia, arXiv, and GitHub.

For the SFT of lightweight models, we deliberately choose commonly adopted datasets with a focus on domain-specific tasks. Furthermore, to fully evaluate the capability of MASA, we employ language understanding and dialogue generation benchmarks. Specifically, the language understanding benchmarks include **BoolQ** (Clark et al., 2019), **HellaSwag** (Zellers et al., 2019), **PIQA** (Bisk et al., 2020), **WinoGrande** (Sakaguchi et al., 2021), **CaseHold** (Zheng et al., 2021), and **MedMCQA** (Pal et al., 2022), while the dialogue generation benchmarks consist of **Dolly**[1], **S-NI** (Wang et al., 2022), **UnNI** (Honovich et al., 2023), **SelfInst** (Wang et al., 2023c), and **VicunaEval** (Chiang et al., 2023).

**Baselines.** To validate the effectiveness of the extracted knowledge, we compare lightweight models inheriting LLM knowledge (MASA) against a randomly initialized baseline (Scratch). In addition, since our approach emphasizes preserving and reusing the generalizable knowledge within LLMs, we also include lightweight baselines (knowledge distillation and pruning) that share a similar goal of retaining critical model knowledge.

We select three types of LLMs as the source models for extracting generalizable knowledge: the dense model OLMo-7B (Groeneveld et al., 2024), the regular MoE model OLMoE-7B (Muennighoff et al., 2025), and the MoE model DeepSeekMoE-16B (Dai et al., 2024) with shared experts. Based on these source models, we establish the following baselines for comparison. **Scratch**: a randomly initialized model without inheriting any knowledge. **Distillation**: a model trained under the teacher-student paradigm, where a larger model guides the pre-training process through knowledge distillation. **Pruning-EEP**: a compact model obtained by applying the experts pruning based method EEP Liu et al. (2024b) to the MoE LLM (OLMoE). **MASA-OLMo**: a lightweight model initialized with knowledge inherited from OLMo-7B. **MASA-OLMoE**: a lightweight model initialized with knowledge inherited from OLMoE-7B. **MASA-DeepSeek**: a lightweight model initialized with knowledge inherited from DeepSeekMoE-16B.

---

[1]https://github.com/databrickslabs/dolly/tree/master

Table 1: Results of lightweight models with different scales on language understanding benchmarks. We experiment with four lightweight models of different sizes. For example, the 12-layer model with hidden size 1024 and FFN intermediate dimension 3072 (12L-267M) corresponds to a 267M-parameter model. All baselines are first pre-trained on 5B tokens data.

| Shape& Params | Baseline | Commonsense & Reading Comprehension | | | | Law | Medicine | Avg. |
|---|---|---|---|---|---|---|---|---|
| | | BoolQ | Hellaswag | PIQA | WinoGrande | CaseHold | MedMCQA | |
| 1024 × 3072 12L-267M | Scratch | 71.65 | 27.15 | 51.31 | 49.80 | 80.20 | 35.07 | 52.53 |
| | Distillation | 69.72 | 26.94 | 51.52 | 48.54 | 81.39 | 32.08 | 51.70 |
| | Pruning-EEP | 65.29 | 24.56 | 47.23 | 48.62 | 34.20 | 31.20 | 41.85 |
| | MASA-OLMo | 72.20 | 27.68 | 51.69 | 50.12 | **83.18** | 35.50 | 53.40 |
| | MASA-OLMoE | 72.91 | 27.27 | 52.39 | 50.20 | 81.90 | 35.62 | 53.38 |
| | MASA-DeepSeek | **73.03** | **27.95** | **55.17** | **51.32** | 83.13 | **35.81** | **54.40** |
| 1024 × 3072 16L-321M | Scratch | 71.53 | 27.11 | 51.09 | 49.88 | 82.62 | 34.57 | 52.80 |
| | Distillation | 71.25 | 27.34 | 51.74 | 48.38 | 82.88 | 34.29 | 52.65 |
| | Pruning-EEP | 65.69 | 24.67 | 48.20 | 48.54 | 51.55 | 28.90 | 44.59 |
| | MASA-OLMo | **73.79** | **27.85** | 57.45 | **50.04** | 83.37 | 36.34 | **54.81** |
| | MASA-OLMoE | 72.84 | 27.39 | **57.51** | 50.04 | 83.52 | 36.27 | 54.60 |
| | MASA-DeepSeek | 72.60 | 27.75 | 54.08 | 49.96 | **84.68** | **36.41** | 54.25 |
| 2048 × 4096 12L-709M | Scratch | 71.35 | 27.06 | 52.61 | 49.88 | 81.05 | 33.21 | 52.53 |
| | Distillation | 70.61 | 27.30 | 50.11 | 48.62 | 79.54 | 34.28 | 51.74 |
| | Pruning-EEP | 59.30 | 25.15 | 51.63 | 48.38 | 52.65 | 27.73 | 44.14 |
| | MASA-OLMo | 73.24 | 28.06 | 53.93 | 50.12 | 82.75 | 35.33 | 53.91 |
| | MASA-OLMoE | 73.67 | 27.91 | **54.30** | 50.20 | **83.62** | 36.34 | 54.34 |
| | MASA-DeepSeek | **73.73** | **29.64** | 53.92 | **50.59** | 82.43 | **36.91** | **54.54** |
| 2048 × 4096 16L-877M | Scratch | 70.61 | 27.92 | 52.34 | 48.78 | 82.33 | 34.99 | 52.83 |
| | Distillation | 71.16 | 27.30 | 51.58 | 49.25 | 81.56 | 32.54 | 52.23 |
| | Pruning-EEP | 57.80 | 25.53 | 51.69 | 48.46 | 52.78 | 27.95 | 44.04 |
| | MASA-OLMo | **73.36** | 30.05 | 54.52 | 50.12 | **83.79** | **36.55** | **54.73** |
| | MASA-OLMoE | 72.05 | **30.14** | 54.73 | 50.28 | 83.73 | 35.93 | 54.48 |
| | MASA-DeepSeek | 73.24 | 28.79 | **56.75** | **50.43** | 83.22 | 35.14 | 54.60 |

Table 2: Results of source LLMs and MASA. The 877M-parameter lightweight model (MASA) is first pre-trained on 10B tokens and then fine-tuned on these SFT datasets. OLMoE-SFT and OLMo-SFT denote the source models that are fine-tuned (full-parameter fine-tuning) until convergence.

| Baseline | #Params | DollyEval | S-NI | UnNI | SelfInst | VicunaEval |
|---|---|---|---|---|---|---|
| OLMo | 7B | 11.18 | 15.82 | 15.16 | 8.68 | 13.38 |
| OLMo-SFT | 7B | 13.27 | 10.81 | 14.06 | 8.34 | 15.39 |
| OLMoE | 7B | 10.78 | 11.10 | 12.25 | 8.46 | 14.42 |
| OLMoE-SFT | 7B | 29.13 | 30.49 | 33.08 | 15.36 | 16.83 |
| MASA | 877M | 25.23 | 19.56 | 24.72 | 11.31 | 14.70 |

**Implementation Details.** MASA-initialized models are dense language models following the Llama architecture, instantiated with varying model dimensions and depths. In our experiments, we use **4M–10M** tokens to train the gene matrices, and **2B–10B** tokens to pre-train the lightweight models. We use the RedPajama-V2 as our pre-training data, and the fine-tuning datasets for downstream tasks are presented in Section 4. During pre-training of the lightweight models, we set the learning rate to $4 \times 10^{-4}$, while in the SFT stage, we use a batch size of 8 and a learning rate of $3 \times 10^{-5}$, with AdamW as the optimizer. For challenging dialogue generation datasets such as DollyEval and S-NI, we follow the configuration of miniLLM and train for 100 epochs. For multiple-choice language understanding datasets such as MMLU and Hellaswag, we train for 3 epochs. We design four target lightweight models with different depths and dimensions, ranging in size from 267M to 877M parameters. To ensure fair comparison, all baselines (e.g., Scratch and Distillation) are configured with the same parameter size for corresponding models, and trained with the same amount of data in both pre-training and SFT stages.

Table 3: Results of models with different scales on dialogue generation benchmarks. All baselines are first pre-trained on 5B tokens data and then fine-tuned on the SFT datasets shown in the table.

| Shape& Params | Baseline | Dialogue Generation | | | | | Avg. (Rouge-L) |
|---|---|---|---|---|---|---|---|
| | | DollyEval | S-NI | UnNI | SelfInst | VicunaEval | |
| 1024 × 3072 12L-267M | Scratch | 21.60 | 14.89 | 19.23 | 8.23 | 12.44 | 15.28 |
| | Distillation | 21.71 | 13.11 | 18.65 | 8.53 | 13.77 | 15.15 |
| | Pruning-EEP | 12.41 | 8.20 | 7.63 | 5.60 | 10.70 | 8.91 |
| | MASA-OLMo | 22.97 | **16.43** | **20.16** | 9.73 | 13.77 | **16.61** |
| | MASA-OLMoE | **23.27** | 16.28 | 19.52 | **9.81** | 13.79 | 16.53 |
| | MASA-DeepSeek | 22.99 | 16.43 | 19.50 | 9.51 | **13.83** | 16.45 |
| 1024 × 3072 16L-321M | Scratch | 22.03 | 14.39 | 17.88 | 8.55 | 14.20 | 15.41 |
| | Distillation | 22.54 | 14.13 | 18.24 | 9.01 | 13.82 | 15.55 |
| | Pruning-EEP | 12.39 | 7.21 | 7.38 | 5.67 | 11.16 | 8.76 |
| | MASA-OLMo | 23.63 | **17.20** | **20.55** | **9.56** | 14.42 | **17.07** |
| | MASA-OLMoE | 22.80 | 15.04 | 19.48 | 9.40 | 14.35 | 16.21 |
| | MASA-DeepSeek | **23.79** | 16.29 | 19.53 | 9.44 | **14.52** | 16.71 |
| 2048 × 4096 12L-709M | Scratch | 22.96 | 15.88 | 20.94 | 9.65 | 14.33 | 16.75 |
| | Distillation | 23.26 | 15.82 | 21.23 | 10.01 | 14.10 | 16.88 |
| | Pruning-EEP | 21.63 | 12.53 | 15.67 | 8.82 | 12.14 | 14.16 |
| | MASA-OLMo | **23.92** | **19.71** | **23.98** | 10.98 | **14.96** | **18.71** |
| | MASA-OLMoE | 23.27 | 16.07 | 21.54 | 10.29 | 14.76 | 17.19 |
| | MASA-DeepSeek | 23.90 | 17.20 | 22.48 | **11.80** | 14.73 | 18.02 |
| 2048 × 4096 16L-877M | Scratch | 23.07 | 17.68 | 22.75 | 10.04 | 14.13 | 17.53 |
| | Distillation | 23.30 | 17.44 | 21.48 | 8.77 | 14.31 | 17.06 |
| | Pruning-EEP | 22.51 | 14.89 | 19.11 | 9.72 | 12.32 | 15.71 |
| | MASA-OLMo | 24.46 | **18.51** | **23.73** | **11.28** | 14.11 | **18.42** |
| | MASA-OLMoE | 24.23 | 18.49 | 22.22 | 10.04 | 14.14 | 17.82 |
| | MASA-DeepSeek | **24.51** | 17.75 | 23.57 | 10.28 | **14.86** | 18.19 |

Figure 2: Comparison of fine-tuning performance between 709M MASA and Scratch under varying pre-training token budgets (2B, 5B, and 10B).

## 4.2 MAIN RESULTS

To comprehensively evaluate the effectiveness of MASA, we conduct experiments from four key perspectives: benchmark performance, comparability to source LLMs, pre-training cost efficiency, and fine-tuning convergence.

**MASA consistently outperforms other initialization methods across most tasks.** As shown in Table 1 and Table 3, MASA surpasses baselines Scratch, Distillation, and Pruning-EEP on nearly all benchmarks (see more results in Appendix A.3). For example, on the S-NI dataset, the 709M MASA-OLMo model outperforms Scratch, Distillation, and Pruning-EEP baselines of the same size by **3.83**, **3.89**, and **7.18**, respectively. We attribute the failures of distillation and pruning to the large capacity gap between source and target models. Previous studies have also shown that excessive teacher–student disparity significantly undermines the effectiveness of knowledge distillation (Mirzadeh et al., 2020; Gao et al., 2020; Wang & Yoon, 2021). Similarly, aggressive pruning ratios can severely damage the structure and functionality of the model, preventing the resulting small model from performing well. In contrast, MASA effectively circumvents these issues by extracting generalizable knowledge from the source LLMs and using it to initialize lightweight models, thereby substantially boosting performance on downstream SFT tasks.

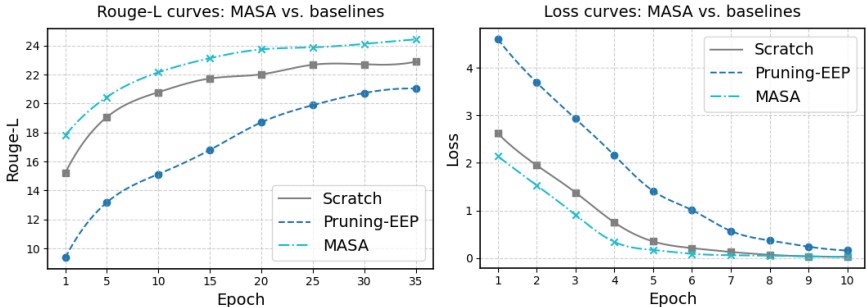

Figure 3: Rouge-L and loss curves on downstream SFT dataset DollyEval.

Table 4: Ablation results for different modules in MASA.

| Method | BoolQ | PIQA | WinoGrande | DollyEval | S-NI | UnNI |
|---|---|---|---|---|---|---|
| w.o. *Spectral Alignment* | 71.77 | 55.44 | 49.96 | 23.38 | 17.35 | 23.16 |
| w.o. *Adaptive Scaling* | 72.14 | 54.84 | 50.20 | 23.58 | 17.20 | 22.24 |
| **MASA** | **73.36** | **56.75** | **50.43** | **24.46** | **18.51** | **23.73** |

**The 877M MASA model, after pre-training on 10B tokens, achieves over 85% of the source LLM's performance on certain SFT datasets.** As shown in Table 2, after inheriting knowledge from OLMoE and undergoing pre-training on 10B tokens, the 877M MASA model achieves **86.6%** and **87.3%** of the 7B source LLM's performance on DollyEval and VicunaEval, respectively. These results demonstrate that MASA effectively transfers the generalizable knowledge of LLMs, enabling lightweight models to inherit broad capabilities and adapt efficiently with limited pre-training.

**MASA requires significantly less pre-training data and offers a cost-efficient training strategy.** As shown in Figure 2, we report results for models pre-trained on 2B, 5B, and 10B tokens before fine-tuning on various SFT datasets (see more results in Appendix A.4). Across all settings, MASA consistently outperforms the Scratch baseline. Notably, on DollyEval, MASA-OLMo pre-trained with only 2B tokens surpasses Scratch pre-trained with 5B tokens, and even approaches the performance of Scratch pre-trained with 10B tokens on UnNI. This demonstrates that MASA can reduce the required pre-training data by **2–5×** on certain datasets, highlighting its ability to transfer generalizable knowledge from LLMs. Such efficiency highlights the validity of the extracted knowledge and confirms its potential for building lightweight models under constrained data budgets.

**MASA converges faster and achieves higher performance on downstream task.** As shown in Figure 3, compared with the Scratch and Pruning baselines, MASA not only reaches better results under the same conditions, but also converges significantly faster on downstream SFT datasets. This accelerated convergence demonstrates that lightweight models initialized with MASA inherit rich generalizable knowledge from large models, enabling them to adapt more efficiently and effectively to diverse tasks. Overall, MASA highlights the potential of leveraging inherited knowledge to improve both training efficiency and generalization in compact models.

### 4.3 ABLATION ANALYSIS

Table 4 presents the ablation results of our dual alignment mechanism and adaptive scaling. In w.o. *Spectral Alignment*, only output alignment is used, which weakens the ability to capture the structural priors of the source model. In w.o. *Adaptive Scaling*, the gene matrix is naively resized by random initialization or direct truncation, leading to knowledge loss during transfer. Both variants yield consistently worse performance than MASA across multiple datasets, demonstrating that spectral alignment is crucial for extracting generalizable knowledge from a principal component perspective, while adaptive scaling ensures flexible dimension matching without disrupting the embedded knowledge structure of the gene matrices. Together, these components are key to fully leveraging source LLM knowledge for effective adaptation.

To validate that the data used for alignment should span various domains, Figure 4 shows the impact of different alignment data on downstream performance. Using single-domain data markedly de-

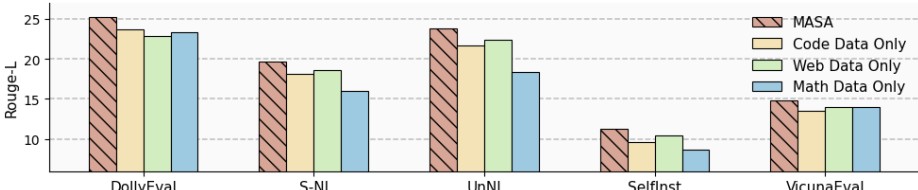

Figure 4: Results of aligning the gene matrices with source LLM parameters using different data types. MASA performs alignment with diverse domains, including but not limited to Wikipedia, ArXiv, and GitHub, ensuring broader knowledge coverage and more effective transfer.

Table 5: Impact of different alignment matrix sizes $M$ on the SFT performance of models. $M/W$ Ratio (%) denotes the proportion of the alignment matrix $M$ relative to the weight matrix of the source LLM. All MASA-initialized models have the same parameter size and are trained with identical amounts of data.

| $M/W$ Ratio (%) | DollyEval | S-NI | UnNI | BoolQ | PIQA | CaseHold | Avg. |
|---|---|---|---|---|---|---|---|
| 3% | 24.67 | 19.00 | 22.22 | 72.14 | 53.21 | 81.90 | 45.52 |
| 12% | 25.24 | 18.91 | 23.32 | 73.36 | 54.73 | 82.75 | 46.38 |
| 23% | 25.23 | 19.56 | 23.87 | 73.61 | 56.75 | 82.43 | 46.91 |
| 40% | 24.25 | 18.51 | 22.13 | 71.47 | 53.26 | 82.06 | 45.28 |

grades MASA's effectiveness, while more diverse data enables the gene matrices to capture broader, task-agnostic knowledge from the source model, thereby enhancing the generalization of lightweight models across tasks.

Moreover, we further investigate how the design choice of gene matrix dimensionality influences the knowledge extraction. Table 5 reports the impact of varying the ratio between the gene matrix and the FFN parameter matrix of source LLM (3%, 12%, 23%, 40%). Performance improves steadily as the ratio increases from 3% to 23%, but shows little or no further gain when expanded to 40%. This indicates that moderately enlarging the gene matrix increases its capacity to capture the dominant spectral structure and generalization-relevant knowledge of the source FFN. However, beyond a certain point, additional capacity begins to absorb lower-energy spectral components that contribute little to generalizable behavior, leading to diminishing returns. Overall, the results suggest that an intermediate ratio provides the best balance between preserving core generalizable patterns and avoiding overfitting to non-essential spectral details.

## 5 CONCLUSION

In this work, we introduced MASA, a unified framework to extract and reuse generalizable knowledge from LLMs. MASA combines matrix-level alignment and scalable adaptation to capture task-agnostic knowledge from FFN layers of LLMs and flexibly transfer it to lightweight models of varying sizes. Extensive experiments across language understanding and dialogue generation tasks in diverse vertical domains demonstrate that MASA consistently improves downstream performance, accelerates convergence, and reduces pre-training data requirements compared to baselines such as random initialization, pruning, and knowledge distillation. These results validate the effectiveness of MASA in extracting and leveraging the generalizable knowledge of LLMs, providing new insights into the structure and utility of generalizable knowledge in large-scale language models.

## ACKNOWLEDGEMENTS

This research was supported by the Jiangsu Science Foundation (BK20243012, BG2024036), the National Science Foundation of China (62125602, U24A20324, 92464301), the New Cornerstone Science Foundation through the XPLORER PRIZE, and the Fundamental Research Funds for the Central Universities (2242025K30024). This work was also supported by the Big Data Computing Center of Southeast University and the Southeast University Ascend Center of Cultivation.

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

# A  APPENDIX

## A.1  SPECTRAL PROPERTIES AND GENERALIZATION

**Theorem A.1** (Spectral Properties and Generalization). *In this section, we provide a theoretical derivation on why the spectral properties of model parameters —in particular the eigenvalue distribution of weight matrices and the Hessian of the loss, are closely related to generalization. Below we outline two complementary perspectives: spectral norm based capacity control and Hessian-based flatness analysis.*

1. ***Generalization Bound via Spectral Norms:*** *Consider an L-layer neural network $f_W$ parameterized by weight matrices $\{W_i\}_{i=1}^L$. Denote the spectral norm by $\|W_i\|_2$ and Frobenius norm by $\|W_i\|_F$. A refined Rademacher complexity bound (Bartlett et al., 2017; Neyshabur et al., 2018) yields that with high probability, the generalization gap*

$$\mathcal{G}(\hat{w}) = \mathbb{E}_{(x,y)\sim\mathcal{D}}[\ell(f_{\hat{w}}(x), y)] - \frac{1}{n}\sum_{j=1}^n \ell(f_{\hat{w}}(x_j), y_j)$$

*satisfies*

$$\mathcal{G}(\hat{w}) \lesssim \frac{B\prod_{i=1}^L \|W_i\|_2}{\sqrt{n}} \sqrt{\sum_{i=1}^L \frac{\|W_i\|_F^2}{\|W_i\|_2^2}}. \tag{7}$$

*Here B is a bound on the input norm. The product $\prod_i \|W_i\|_2$ upper bounds the Lipschitz constant of the network, while the ratio $\|W_i\|_F^2/\|W_i\|_2^2$ reflects the effective dimensionality of each layer. Equation equation 7 explicitly demonstrates that controlling spectral norms improves generalization guarantees.*

2. ***Flatness via Hessian Spectrum:*** *Let $w^\star$ denote a (local) minimizer of the training loss $\mathcal{L}(w)$. A second-order Taylor expansion around $w^\star$ gives*

$$\mathcal{L}(w^\star + \varepsilon) \approx \mathcal{L}(w^\star) + \tfrac{1}{2}\varepsilon^\top H \varepsilon, \tag{8}$$

*where $H = \nabla_w^2 \mathcal{L}(w^\star)$ is the Hessian. Assuming $\varepsilon \sim \mathcal{N}(0, \sigma^2 I)$, we obtain*

$$\mathbb{E}_\varepsilon[\mathcal{L}(w^\star + \varepsilon)] - \mathcal{L}(w^\star) \approx \tfrac{1}{2}\sigma^2 \operatorname{Tr}(H). \tag{9}$$

*Since $\operatorname{Tr}(H)$ is the sum of eigenvalues of $H$, Equation equation 9 shows that large eigenvalues imply higher sensitivity to parameter perturbations, i.e. a sharper minimum. PAC-Bayes analyses (Dziugaite & Roy, 2017; Neyshabur et al., 2017) further formalize this connection, establishing that flatter solutions (smaller Hessian spectrum) yield tighter generalization bounds.*

SUMMARY

Equations equation 7 and equation 9 illustrate two distinct but related spectral characterizations:

- The **spectral norm** of weight matrices controls network Lipschitz constant and capacity, directly entering Rademacher-based generalization bounds.
- The **Hessian spectrum** controls loss sensitivity to parameter perturbations, and via PAC-Bayes or stability arguments, connects to generalization error.

Although these bounds are often loose, they provide a principled explanation of the empirical observation that solutions with smaller spectral norms and flatter Hessian spectra typically generalize better. Therefore, in our method, employing spectral alignment is crucial for extracting the generalizable knowledge from large models.

## A.2 DETAILED COMPRESSION AND DECOMPRESSION PROCESS IN THE DUAL ALIGNMENT MECHANISM

In the output alignment of the dual alignment mechanism (Section 3.1), the input vector $x$ and the gene matrix $M$ generally have mismatched dimensions. To address this, we introduce a compression–decompression process for $x$. Specifically, given an input vector $x \in \mathbb{R}^{d_{in}}$, we first reduce its dimensionality to $r$, which matches the dimension of the gene matrix $M \in \mathbb{R}^{r \times r}$. We reshape $x$ into $n$ sub-vectors of length $r$:

$$n = \lceil \frac{d_{in}}{r} \rceil, x \to x_c \in \mathbb{R}^{n \times r}. \tag{10}$$

If $d_{in}$ is not divisible by $r$, we apply padding to the last sub-vector by duplicating elements from the tail of $x$, ensuring $x_c$ has exactly $nr$ entries.

Each sub-vector $x_c^{(i)} \in \mathbb{R}^r$ is then transformed independently by the gene matrix:

$$z^{(i)} = M x_c^{(i)}, \quad z^{(i)} \in \mathbb{R}^r. \tag{11}$$

We then concatenate all transformed blocks along the last dimension:

$$z = \operatorname{concat}(z^{(1)}, z^{(2)}, \ldots, z^{(n)}) \in \mathbb{R}^{nr}. \tag{12}$$

Finally, to match the target dimension $d_{out}$, we adjust $z$ as follows:

- If $nr > d_{out}$, we truncate to the first $d_{out}$ dimensions:

$$\tilde{x} = z[: d_{out}] \in \mathbb{R}^{d_{out}}. \tag{13}$$

- If $nr \leq d_{out}$, we repeat or pad $z$ until its length exceeds $d_{out}$, and then truncate:

$$\tilde{x} = \operatorname{repeat}(z, \lceil \frac{d_{out}}{(nr)} \rceil)[: d_{out}] \in \mathbb{R}^{d_{out}}. \tag{14}$$

This compression–decompression process enables dimension-consistent alignment between the input vector $x$ and the gene matrix $M$, ensuring that knowledge transfer can be performed seamlessly across different model scales.

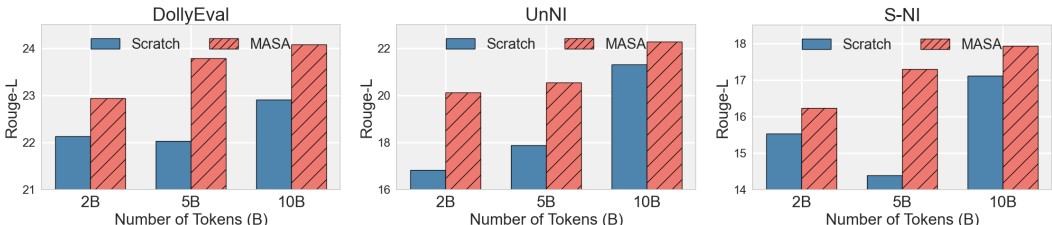

Figure 5: Comparison of fine-tuning performance between 321M MASA and Scratch under varying pre-training token budgets (2B, 5B, and 10B).

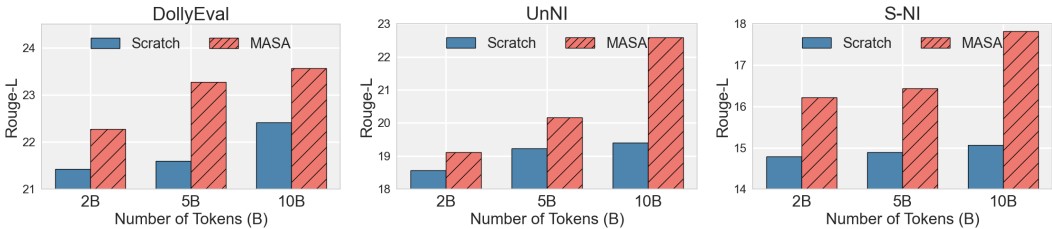

Figure 6: Comparison of fine-tuning performance between 267M MASA and Scratch under varying pre-training token budgets (2B, 5B, and 10B).

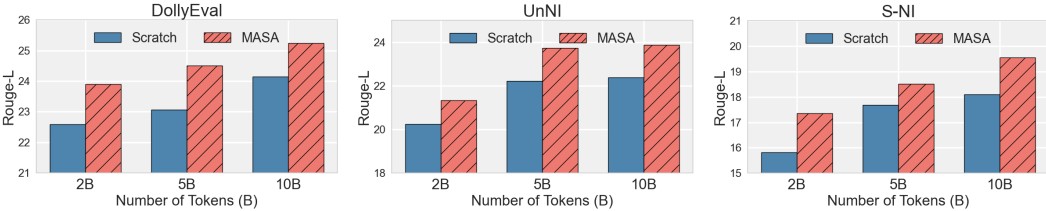

Figure 7: Comparison of fine-tuning performance between 877M MASA and Scratch under varying pre-training token budgets (2B, 5B, and 10B).

### A.3 MORE RESULTS WITH DIFFERENT AMOUNT OF PRE-TRAINED DATA

To thoroughly demonstrate the effectiveness of our approach, we compare MASA against multiple baselines under different amounts of pre-training data, followed by fine-tuning on SFT datasets. Specifically, we report baseline performance on dialogue generation datasets after pre-training with 2B and 10B tokens, as well as performance on language understanding datasets after pre-training with 2B tokens. As shown in the tables (Table 6, Table 7 and Table 8), regardless of the amount of pre-training data, our method consistently outperforms the baselines across the majority of benchmarks. These results highlight that MASA not only provides a more efficient initialization strategy but also enables lightweight models to achieve stronger generalization and task adaptability, even under limited pre-training data.

### A.4 DATA EFFICIENCY ACROSS DIFFERENT MODEL SIZES

In this section, we present the histogram results of Scratch and MASA baselines across different model sizes (267M, 321M and 877M), after pre-training on 2B, 5B, and 10B tokens, followed by SFT on DollyEval, UnNI, and S-NI datasets. As shown in Figure 5 6 7, the results show that MASA consistently outperforms the Scratch baseline across all settings, regardless of the amount of pre-training data. Remarkably, on certain datasets, a MASA model pre-trained with only 2B tokens even surpasses the performance of a Scratch model pre-trained with 10B tokens. These findings demonstrate that the knowledge extracted from the source LLM can substantially enhance the performance of lightweight models, achieving up to a $5\times$ reduction in pre-training data requirements while maintaining or even improving effectiveness.

### A.5 THEORETICAL FOUNDATIONS FOR FFN-BASED KNOWLEDGE EXTRACTION

We focus on the Feed-Forward Network (FFN) layers as the foundation for extracting generalizable knowledge from LLMs. FFN layers account for the majority of model parameters and have been

Table 6: Results of lightweight models with different scales on language understanding benchmarks. We experiment with four lightweight models of different sizes. For example, the 12-layer model with hidden size 1024 and FFN intermediate dimension 3072 (12L-267M) corresponds to a 267M-parameter model. All baselines are first pre-trained on **2B tokens** data.

| Shape& Params | Baseline | Commonsense & Reading Comprehension | | | | Law | Medicine | Avg. |
|---|---|---|---|---|---|---|---|---|
| | | BoolQ | Hellaswag | PIQA | WinoGrande | CaseHold | MedMCQA | |
| 1024 × 3072 12L-267M | Scratch | 70.00 | 25.77 | 50.27 | 48.54 | 78.33 | 33.8 | 51.12 |
| | Distillation | 69.94 | 25.08 | 50.98 | 48.38 | 80.33 | 31.46 | 51.03 |
| | MASA-OLMo | 70.28 | **26.81** | 50.82 | 49.80 | 81.35 | **35.41** | 52.41 |
| | MASA-OLMoE | **70.55** | 26.18 | 50.92 | 49.88 | 80.95 | 34.71 | 52.20 |
| | MASA-DeepSeek | 70.40 | 26.48 | **51.25** | **50.17** | **81.59** | 34.93 | **52.47** |
| 1024 × 3072 16L-321M | Scratch | 70.03 | 26.13 | 51.09 | 48.62 | 80.52 | 34.37 | 51.79 |
| | Distillation | 70.64 | 26.25 | 51.31 | 48.30 | 82.37 | 34.05 | 52.15 |
| | MASA-OLMo | 70.70 | 26.49 | 52.23 | 48.78 | **83.14** | 34.95 | 52.72 |
| | MASA-OLMoE | 70.49 | 26.79 | **52.99** | **48.93** | 82.65 | 34.90 | 52.79 |
| | MASA-DeepSeek | **70.76** | **26.92** | 52.88 | 48.86 | 83.11 | **35.24** | **52.96** |
| 2048 × 4096 12L-709M | Scratch | 70.09 | 26.88 | 52.72 | 48.70 | 79.80 | 31.58 | 51.63 |
| | Distillation | 69.24 | 26.76 | 47.77 | 48.38 | 78.80 | 33.40 | 50.73 |
| | MASA-OLMo | **71.83** | **28.21** | 53.21 | 48.85 | 81.08 | 35.05 | 53.04 |
| | MASA-OLMoE | 71.25 | 27.81 | **54.08** | 48.93 | **81.95** | 34.69 | **53.12** |
| | MASA-DeepSeek | 71.64 | 27.93 | 52.50 | **49.33** | 80.99 | **36.19** | 53.10 |
| 2048 × 4096 16L-877M | Scratch | 68.41 | 26.78 | 52.62 | 48.70 | 81.16 | 34.40 | 52.01 |
| | Distillation | 70.28 | 27.04 | 51.41 | 48.78 | 80.56 | 33.68 | 51.96 |
| | MASA-OLMo | **72.48** | **28.39** | 54.03 | 49.96 | 83.01 | 35.24 | 53.85 |
| | MASA-OLMoE | 70.92 | 28.25 | 54.35 | **50.12** | 82.31 | **36.89** | 53.81 |
| | MASA-DeepSeek | 72.05 | 28.11 | **56.09** | 49.33 | **83.13** | 34.86 | **53.93** |

shown to serve as key-value memories storing factual and conceptual knowledge (Geva et al., 2020; Meng et al., 2022). Unlike attention layers, which mainly capture token interactions, FFN layers encode token-specific representations that are more stable across tasks, making them suitable for extracting domain-agnostic knowledge. Moreover, previous studies reveal that FFN weights exhibit strong low-rank structures (Aghajanyan et al., 2020; Hu et al., 2022), suggesting redundancy and enabling compact representation learning. These properties make FFN layers an ideal target for extracting generalizable knowledge modules that can be effectively transferred to smaller models.

## A.6 THE USE OF LLMS

In this paper, we only employ large language models (LLMs) for grammatical correction and language polishing of the manuscript. The use of LLMs does not involve the core ideas, methodological design, experimental implementation, or result analysis of the paper. We affirm that all methods and code, as well as the important equations and figures presented in the paper, are entirely written and produced by the authors. We declare that all the generated content of LLMs has been reviewed by the authors.

Table 7: Results of models with different scales on dialogue generation benchmarks. All baselines are first pre-trained on **2B tokens** data and then fine-tuned on the SFT datasets shown in the table.

| Shape& Params | Baseline | Dialogue Generation | | | | | Avg. (Rouge-L) |
|---|---|---|---|---|---|---|---|
| | | DollyEval | S-NI | UnNI | SelfInst | VicunaEval | |
| 1024 × 3072 12L-267M | Scratch | 21.43 | 15.06 | 18.56 | 9.07 | 13.34 | 15.49 |
| | Distillation | 21.54 | 13.41 | 17.41 | 8.30 | 14.10 | 14.95 |
| | MASA-OLMo | 22.17 | **16.22** | **19.12** | 9.54 | **14.39** | **16.29** |
| | MASA-OLMoE | **22.27** | 14.59 | 18.95 | **9.74** | 14.37 | 15.98 |
| | MASA-DeepSeek | 21.83 | 15.89 | 18.79 | 9.47 | 14.21 | 16.04 |
| 1024 × 3072 16L-321M | Scratch | 22.13 | 15.54 | 16.84 | 8.87 | 14.20 | 15.52 |
| | Distillation | 21.10 | 14.13 | 18.24 | 9.01 | 13.59 | 15.21 |
| | MASA-OLMo | **22.94** | **16.23** | **20.13** | 9.43 | 14.32 | **16.61** |
| | MASA-OLMoE | 22.55 | 14.47 | 18.77 | 9.16 | **14.51** | 15.89 |
| | MASA-DeepSeek | 22.54 | 16.09 | 19.44 | **9.65** | 14.06 | 16.36 |
| 2048 × 4096 12L-709M | Scratch | 22.49 | 15.63 | 20.43 | 8.58 | 13.64 | 16.15 |
| | Distillation | 22.88 | 15.14 | 19.20 | 8.98 | 13.59 | 16.88 |
| | Pruning-EEP | 21.63 | 12.53 | 15.67 | 8.82 | 12.14 | 15.96 |
| | MASA-OLMo | 23.14 | 16.92 | 20.44 | **10.05** | **14.40** | **16.99** |
| | MASA-OLMoE | 23.29 | **16.99** | **20.66** | 9.42 | 13.30 | 16.73 |
| | MASA-DeepSeek | **23.31** | 16.56 | 20.02 | 9.93 | 13.91 | 16.75 |
| 2048 × 4096 16L-877M | Scratch | 22.59 | 15.82 | 20.24 | 9.17 | 13.94 | 16.35 |
| | Distillation | 22.25 | 16.29 | 20.21 | 8.67 | 13.74 | 16.23 |
| | MASA-OLMo | 23.75 | 16.71 | 20.18 | 9.93 | 14.58 | 17.03 |
| | MASA-OLMoE | 23.67 | **17.35** | **21.34** | 10.19 | **14.89** | **17.49** |
| | MASA-DeepSeek | **23.90** | 16.45 | 20.55 | **10.29** | 14.41 | 17.12 |

Table 8: Results of models with different scales on dialogue generation benchmarks. All baselines are first pre-trained on **10B tokens** data and then fine-tuned on the SFT datasets shown in the table.

| Shape& Params | Baseline | Dialogue Generation | | | | | Avg. (Rouge-L) |
|---|---|---|---|---|---|---|---|
| | | DollyEval | S-NI | UnNI | SelfInst | VicunaEval | |
| 1024 × 3072 12L-267M | Scratch | 22.41 | 14.79 | 19.4 | 8.86 | 13.54 | 15.80 |
| | Distillation | 21.87 | 14.79 | 19.55 | 8.99 | 13.82 | 15.80 |
| | MASA-OLMo | 23.50 | **17.82** | **22.59** | 10.40 | **15.00** | **17.86** |
| | MASA-OLMoE | **23.57** | 17.79 | 21.94 | **10.42** | 14.20 | 17.58 |
| | MASA-DeepSeek | 23.02 | 16.44 | 20.85 | 10.04 | 14.14 | 16.90 |
| 1024 × 3072 16L-321M | Scratch | 22.91 | 17.11 | 21.32 | 10.04 | 14.24 | 17.12 |
| | Distillation | 22.45 | 14.85 | 18.63 | 9.11 | 13.64 | 15.74 |
| | MASA-OLMo | **24.08** | 16.42 | 21.02 | 10.12 | 14.59 | 17.25 |
| | MASA-OLMoE | 23.63 | 17.04 | 21.87 | **10.51** | **14.76** | 17.56 |
| | MASA-DeepSeek | 23.89 | **17.94** | **22.28** | 10.37 | 13.95 | **17.69** |
| 2048 × 4096 12L-709M | Scratch | 23.99 | 19.10 | 21.23 | 9.68 | 13.84 | 17.57 |
| | Distillation | 23.38 | 17.48 | 22.56 | 10.16 | 13.04 | 17.32 |
| | MASA-OLMo | **24.95** | 19.76 | 23.45 | 10.11 | 14.49 | 18.55 |
| | MASA-OLMoE | 24.73 | **20.18** | **24.94** | **10.40** | 14.59 | **18.97** |
| | MASA-DeepSeek | 24.51 | 19.95 | 23.32 | 10.32 | **14.78** | 18.58 |
| 2048 × 4096 16L-877M | Scratch | 24.14 | 18.10 | 22.38 | 10.22 | 14.16 | 17.80 |
| | Distillation | 23.65 | 17.79 | 23.21 | 9.55 | 14.11 | 17.66 |
| | MASA-OLMo | 25.23 | 19.00 | 23.87 | 10.39 | 14.37 | 18.57 |
| | MASA-OLMoE | **25.24** | 18.91 | **24.72** | 10.38 | 14.18 | **18.69** |
| | MASA-DeepSeek | 24.67 | **19.56** | 23.45 | **10.72** | **14.70** | 18.62 |

