# OpenReview forum: "Inheriting Generalizable Knowledge from LLMs to Diverse Vertical Tasks"
_ICLR.cc/2026/Conference — ICLR 2026 Poster_

### Official Review · Reviewer_KeEV · 2025-10-24

**Soundness:** 2
**Presentation:** 3
**Contribution:** 3
**Rating:** 4
**Confidence:** 4

**Summary:**

This paper proposes a technique to extract information stored in the feedforward layers from a pretrained LLM and uses it to initialize a smaller LLM of arbitrary dimension. The key idea consists of two steps: 1) extract square matrices from the FFN weight matrices of the LLM using a combination of output alignment that minimizes the distance between the outputs from the original layers and the square matrices and spectral alignment that minimizes the distance between singular values of the original and square matrices, and 2) adaptive scaling mechanism which reshapes the extracted square matrices and uses it to initialize a smaller LLM of arbitrary dimension. The remaining parameters of the smaller LLM are randomly initialized.  The paper applies the method to 3 different source LLMs: OLMo-7B, OLMoE-7B and DeepSeekMoE-16B. The paper shows that the proposed method can yield improvements in performance when compared to a number of baselines including training from scratch, distillation from a larger model and pruning. This is true for models of different sizes. The technique achieves 85% of the performance obtainable via full fine-tuning. It requires 2-5x less pre-training data when compared to training from scratch. It converges faster when fine-tuning on specific datasets. The paper reports an ablation analysis showing that both components of  the model - spectral alignment and adaptive scaling - play a critical role.

**Strengths:**

* Proposes a novel technique to extract useful information from a pre-trained LLM which can used to initialize a smaller model of arbitrary dimension
* Shows that the technique improves upon alternative initialization approaches such as training from scratch, distillation and pruning.
* Shows that the method can reduce the amount of data for pre-training by a factor of 2-5x, and can reduce convergence time in fin-tuning.

**Weaknesses:**

* The evaluation contains some baselines which are not comparable. See questions below.
* The paper does not provide some details. See questions below.

**Questions:**

* L016: The abstract contains the term 'gene matrices' which are previously undefined.
* L170: Is the number of square matrices identical to the number of weight matrices in the LLM?
* Evaluation: In Tables 1 and 2, what is the larger model used for Distillation? For proper comparison, the model for Distillation and Pruning-EEP should be identical to that used for MASA.  At the moment, a) The teacher model for Distillation is not specified and b) Pruning-EEP is based on OLMoE. Thus, it seems like Pruning-EEP can not be compared to MASA-OLMo or MASA-DeepSeek. Please clarify.
* Evaluation: For Table 4, the paper states that "In w.o Adaptive Scaling, the gene matrix is naively resized by random initialization or direct truncation". When is random initialization chosen over truncation? or is the decision random?
* Figure 4: What are the proportion of the different datasets : code/web/math?
* L175: "gene matrices are aligned with experts that are highly active across diverse tasks" How do you quantify this?
* L202: f_c(x_in), the x_in looks like a subscript.

Typos:
* L071: cross -> across

---

> ### Author Response · Authors · 2025-11-20
> **Response to Reviewer #KeEV**
>
> Thank you for your careful and thorough review of our work and for all the corrections you have pointed out. We have addressed all the issues and uploaded the revised version of the manuscript. Regardless of your final evaluation of our paper, we sincerely appreciate your professional attitude and the valuable questions and suggestions you have provided. We wish you all the best in your research.
>
> ***Q1***   **Is the number of square matrices identical to the number of weight matrices in the LLM?**
>
> ***R1*** Yes. The learngene matrices correspond **one-to-one** with the weight parameters in the FFN layers of the large model. In other words, for each FFN layer, the number of learngene matrices matches exactly the number of weight matrices in that layer.
>
> ***Q2***   **Evaluation: In Tables 1 and 2, what is the larger model used for Distillation?**
>
> ***R2*** For the distillation baseline used in our comparison, we adopt the **MiniLLM** distillation framework. As for the ​**teacher model**​, we follow the results reported in the **OLMoE** paper. Since **OLMoE-7B** consistently outperforms **DeepSeekMoE-16B** on downstream tasks, we select the stronger **OLMoE-7B** as the teacher model for our distillation baseline.
>
> ***Q3***   **Evaluation: For Table 4, the paper states that "In w.o Adaptive Scaling, the gene matrix is naively resized by random initialization or direct truncation". When is random initialization chosen over truncation? or is the decision random?**
>
> ***R3*** In the **w.o Adaptive Scaling** baseline, we directly crop or resize the matrices to match the target dimensions **without considering the original structural properties or functional roles** of the weights. This baseline is included to provide a clear comparison against our **Adaptive Scaling** method, which explicitly preserves and adjusts the internal knowledge structure of the matrices during resizing.
>
> ***Q4***  **Figure 4: What are the proportion of the different datasets : code/web/math?**
>
> ***R4***  Thank you for your question. In Figure 4, to ensure fairness in comparison, the amounts of data used for different domains (such as code, web, and math) are kept ​**equal**​. Likewise, for the MASA model trained on mixed-domain data, the proportions of samples from each domain are also kept ​**nearly identical**​, ensuring that no single domain dominates the training process.
>
> ***Q5***   **L175: "gene matrices are aligned with experts that are highly active across diverse tasks" How do you quantify this?**
>
> ***R5***  Thank you very much for your thorough review and insightful questions. For MoE architectures that **do not explicitly designate shared experts** (such as OLMoE), our approach is as follows:
>
> We feed data from **multiple different domains** into the model and carefully analyze the activation patterns of all experts. By examining how each expert responds across these diverse domains, we identify a subset of experts that exhibit **consistently high activation** regardless of the input domain.
>
> These highly responsive experts effectively serve as ​**implicit shared experts**​—they contain knowledge that is **more generalized and less task-specific** compared to other experts. Such experts naturally align with our goal of extracting ​**cross-domain, generalizable knowledge**​, making them suitable candidates for constructing the learngenes.
>
> ***Q6***   **L202: f\_c(x\_in), the x\_in looks like a subscript.**
>
> ***R6*** Thank you for your careful and thorough review of our work and for all the corrections you have pointed out. We have addressed all the issues and uploaded the revised version of the manuscript.

---

> > ### Comment · Reviewer_KeEV · 2025-11-25
> >
> > Thank you for your responses. They answer many of my questions. I still have one more question:
> >
> > * For "Q5 L175: "gene matrices are aligned with experts that are highly active across diverse tasks" How do you quantify this", you mention that "we identify a subset of experts that exhibit consistently high activation regardless of the input domain". How do define 'consistently high' ? What is the metric?

---

> > > ### Author Response · Authors · 2025-11-26
> > > **Response to Reviewer #KeEV**
> > >
> > > Thank you for your response. Specifically, in selecting experts, we feed data from different domains (such as wekipedia data, arXiv data, GitHub data, etc) into the model and record the routing patterns of experts at each layer. Based on the expert selections for tokens from each domain, we compute the activation frequency of every expert across domains, as well as the entropy of these activation frequencies. Experts with consistently high activation across all domains (i.e., those with low entropy) are identified as “common experts” and chosen as the targets for our matrix alignment.
> > >
> > > In the current manuscript, we report results obtained by aligning to the most activated expert. In future work, we plan to explore more sophisticated expert selection strategies and investigate aligning multiple experts to obtain a more comprehensive understanding of shared knowledge.

---

### Official Review · Reviewer_Nd6r · 2025-11-01

**Soundness:** 3
**Presentation:** 3
**Contribution:** 2
**Rating:** 4
**Confidence:** 3

**Summary:**

This paper proposes MASA, a method to extract "generalizable knowledge" from large language models (LLMs) via compact "gene matrices" aligned with Feed-Forward Network (FFN) layers using output and spectral alignment. An adaptive scaling technique is introduced to transfer these matrices to initialize smaller models. Experiments across multiple LLM architectures and tasks show that MASA-initialized models outperform random initialization, distillation, and pruning baselines in performance, data efficiency, and convergence speed.

**Strengths:**

- This paper presents a method for transferring knowledge from large to small LLMs.
- Experiments cover multiple model architectures (Dense, MoE) and a range of downstream tasks.
- The paper is well-written and easy to follow.

**Weaknesses:**

- The core concept lacks strong novelty due to its significant overlap with the Learngene framework.
- Insufficient evaluation of "generalizable knowledge". Most evaluations are on supervised fine-tuning tasks. Testing in zero-shot or few-shot settings would better demonstrate the transfer of generalizable knowledge.
- The paper lacks of sensitivity study on the hyperparameters λ.

**Questions:**

- Given the Learngene framework already established the core principle of inheriting condensed knowledge, what is the fundamental conceptual advance of MASA beyond its application to LLMs? Can you provide a detailed comparison?
- To substantiate the claim of transferring "generalizable knowledge," can you provide performance on standard zero-shot benchmarks for models initialized with MASA before any task-specific fine-tuning?
- The spectral alignment loss uses a fixed hyperparameter λ. Can you provide a sensitivity analysis to show how critical this parameter is to the final performance?

---

> ### Author Response · Authors · 2025-11-20
> **Response to Reviewer #Nd6r**
>
> Thank you for your careful reading and insightful questions.
>
> ***Q1***   **what is the fundamental conceptual advance of MASA beyond its application to LLMs? Can you provide a detailed comparison?**
>
> ***R1*** Thank you for your question. The proposed approach of extracting **generalizable knowledge** from large models through a **learngene matrix** has indeed not been explored before. Even within the Learngene framework in the computer vision community, prior work typically **directly extracts parameters** from the model as learngenes, without attempting to use an intermediate representation or transformation as we do.
> Therefore, our method, leveraging a mediating learngene matrix to extract and transfer shared knowledge, is genuinely novel and represents an original contribution beyond existing Learngene-based techniques.
>
> ***Q2***   **can you provide performance on standard zero-shot benchmarks for models initialized with MASA before any task-specific fine-tuning?**
>
> ***R2*** Thank you for the suggestion. We agree that reporting zero-shot results can provide additional context. However, our method focuses on ​**initialization for efficient adaptation**​, rather than improving inherent zero-shot capability.
> It is important to note that our model has only ​**570M parameters**​, and models of this scale—regardless of the initialization method—typically exhibit ​**very limited zero-shot performance**​. This is a well-known limitation of small language models: strong zero-shot reasoning generally emerges only at the multi-billion-parameter scale.
> Therefore, comparing zero-shot performance between a 570M model and much larger teacher models (such as 7B or 16B) would not be meaningful and would primarily reflect ​**model capacity differences rather than the quality of the initialization method**​.
>
> ***Q3***   **The spectral alignment loss uses a fixed hyperparameter λ**
>
> ***R3*** Thank you very much for your suggestion. We conducted ablation experiments under different λ values and provided the average results on the dataset. We can observe that when λ is either too large or too small, the alignment quality of the gene matrix deteriorates. The best results are obtained when λ is in the range of 0.2–0.5.
>
> | Value of  $\lambda$ | Dolly | Boolq | PIQA | AVG|
> | --- | --- | --- | --- | --- |
> | 0 | 23.38 | 71.77 | 55.44| 50.20
> | 0.05 | 23.06 | 71.89 | 55.31| 50.09
> | 0.2 | 24.46 | 73.36 | 56.75 | 51.52
> | 0.5 |24.08 | 73.79 | 57.45 | 51.77
> |0.8 | 22.94 | 71.53 | 54.08 | 49.52

---

### Official Review · Reviewer_Wu8B · 2025-11-03

**Soundness:** 2
**Presentation:** 2
**Contribution:** 2
**Rating:** 2
**Confidence:** 5

**Summary:**

This paper proposed a method to extract knowledge from source LLMs and inherit the knowledge to lightweight models for downstream adaptation. Specifically, in knowledge extraction, the paper applied output reconstruction loss and singular value matching loss to make sure the gene matrices can extract the useful information from the FFN layers in the source LLM; to apply the gene matrices to lightweight models, the paper applied matrix column and row expansion based on the norm value of the left and right singular matrices. The proposed method was compared to other approaches including model distillation and pruning, and showed better task metrics on understanding and generative tasks.

**Strengths:**

This paper is a fusion of different important aspects in LLM adaptation, including (1) how to distill large model to lightweight ones for better efficiency, (2) how to extract knowledge from source LLM to downstream adaptation for better accuracy, (3) how to address potential weight matrix mismatch problem between source and target LLM architectures.

**Weaknesses:**

I have concerns in the following perspectives:
- Novelty. In the knowledge extraction section, the paper proposed a method to align the output from the source LLM and the gene metrics based on MSE loss. These type of reconstruction objective has been well established in the deep neural network pruning community. Just list a few papers as reference [1] [2] [3]. Please add proper references to these previous works.
- Technical clarity. (1) The theoretical justification about the single value alignment loss in the knowledge extraction section is not clear. Based on the provided justification in the appendix, the generalization is related to the flatness of the Hessian matrix, which is affected by the Trace of the Hessian matrix, which is the sum of eigen values of the Hessian matrix, but not the FFN weight matrices. Even we want to use the Rademacher bound to showcase that the spectral norms of the weight matrices affect generalization, based on the bound, I would say it is making more sense to create loss function based on minimizing the discrepancy of the Frobenums norm of the source weight matrix and gene matrix, or minimize the discrepancy of the nuclear norm of two matrices. The paper did not provide convincing ablation study. (2) In the knowledge inheriting section, the authors seem want to use some of the concept from matrix column set selection and importance score to determine how to expand the rows and columns of the gene matrix. However, based on the important score sampling in column subset selection theory, we should actually sample the original weight matrix's column based on the right singular weight matrix row-wise norm (i.e., importance score). The current implementation does not have clear theoretical justification.
- Experiments. The results section misses an important baseline, which is pruning + SFT with distillation. The proposed approach extracts existing knowledge from source LLM, then apply to target LLM as initialization, and then SFT. Moreover, the two alignment loss acts like distillation to make sure the gene matrices can mimic the source LLM. Therefore, it is not suprising that the approach is better than training from scratch, distillation, or pruning. The real baseline should be applying pruning to get better initialization from the source LLM, apply SFT or other finetuning strategy with distillation loss so that the target LLM can mimic the source LLM, similar to the dual alignment loss proposed in the paper. Without this baseline, it is hard to make judgement on the real effectiveness of the approach.

[1]  Channel pruning for accelerating very deep neural networks. ICCV 2017
[2] Thinet: A filter level pruning method for deep neural network compression. ICCV 2017
[3] Discrimination-aware channel pruning for deep neural networks. Neurips 2018

**Questions:**

Please refer to the weakness section.

---

> ### Author Response · Authors · 2025-11-20
> **Response to Reviewer #Wu8B**
>
> Thank you for your careful reading and insightful questions.
>
> ***Q1***   **Novelty.**
>
> ***R1***  Thanks for your suggestion. Although we also employ MSE to align the output of the source model, our method fundamentally differs from its use in pruning.
> Our goal is ​**not to compress the model**​, but rather to ​**extract cross-task shared knowledge**​.
> We do **not perform any pruning or structural reduction** on the source model; instead, we design a learnable matrix to selectively distill and extract its generalizable knowledge.
> Moreover, the alignment process in our approach is ​**not limited to MSE**​—we also incorporate ​**spectral alignment**​, making the extraction more robust and semantically meaningful compared to traditional reconstruction-based methods.
>
> ***Q2***   **Technical clarity.**
>
> ***R2***  (1) Thank you for the insightful comment. Our single value alignment loss is ​**not intended to reconstruct the full weight matrix**​, but rather to **align the dominant spectral components** that encode the shared structural knowledge across experts. We agree that the explanation in the appendix may overemphasize the Hessian trace argument, and we will revise the discussion to focus on ​**norm-based capacity measures**​, where dominant singular values directly influence model complexity and generalization.
>
> While Frobenius or nuclear norm discrepancies are valid alternatives, they impose ​**full-matrix constraints**​, which we found unnecessary for extracting transferable low-rank structure in practice. We appreciate the suggestion and will clarify this rationale in the revision, along with plans to explore such norms in future work.
>
> (2) Thank you for the comment. Our method for expanding the gene matrix does follow the theoretical foundation of column subset selection. Specifically, we compute the importance score using the **column-wise** L_2 norm of the right singular matrix V. This is a standard and theoretically justified criterion for measuring column importance, as it directly relates to the contribution of each column to the low-rank structure of the original matrix [e.g., “CUR Decomposition”, Mahoney & Drineas, PNAS'09].
>
> We will clarify this in the manuscript to avoid ambiguity. Thank you for pointing this out.
>
> ***Q3***   **Experiments.**
>
> ***R3*** Thank you for the insightful suggestion. We agree that combining pruning with fine-tuning and distillation could serve as an alternative way to initialize smaller models. However, our pruning baseline is designed to represent approaches ​**that do not rely on pretraining or post-distillation**​, and therefore matches the purpose of comparing effective initialization strategies under limited resources.
>
> Incorporating both pruning and distillation, as suggested, would constitute a ​**hybrid approach combining structure compression and knowledge transfer**​, which we view as orthogonal and potentially compatible with our GeneLLM framework. While valuable, such pipelines require substantial additional experiments and are beyond the scope of this work. We appreciate this suggestion and consider it a promising direction for future studies.

---

### Official Review · Reviewer_yUAD · 2025-11-04

**Soundness:** 2
**Presentation:** 3
**Contribution:** 2
**Rating:** 4
**Confidence:** 4

**Summary:**

This paper proposes MASA (Matrix-level Alignment and Scalable Adaptation), a two-stage framework to extract generalizable knowledge from large language models (LLMs) and use it to initialize smaller, lightweight models.

- Knowledge Extraction: The method trains a lightweight set of "gene matrices" to capture the knowledge within the FFN layers of a frozen source LLM. This training uses a dual alignment loss: a standard output alignment loss ($L_{out}$) combined with a novel spectral alignment loss ($L_{spec}$). $L_{spec}$ aims to match the singular value distribution of the gene matrix to that of the source FFN weight matrix.
- Knowledge Inheriting: The framework uses a scalable adaptation strategy to transfer the trained gene matrices to target models of arbitrary FFN dimensions. This SVD-based method trims or pads the singular vector matrices to match the target dimensions before reconstruction.

Experiments show that models initialized with MASA converge faster and achieve stronger performance compared to baselines like random initialization, pruning, and knowledge distillation.

**Strengths:**

- Practical Problem: The paper addresses the significant and practical problem of efficiently transferring the vast knowledge of LLMs to smaller, more deployable models.
- Technically Motivated: The methodology includes non-trivial technical ideas. The spectral alignment loss ($L_{spec}$) is a principled addition, moving beyond simple functional imitation (like in distillation) to match deeper structural properties of the weight matrices, with good theoretical motivation.

- Principled Scaling: The SVD-based "scalable adaptation" is a principled and flexible method for transferring parameters across models of different FFN dimensions, superior to naive truncation.

**Weaknesses:**

- Missing Baseline: It compares MASA against "random initialization, pruning, and knowledge distillation" but fails to include  relevant class of baselines: direct parameter transfer. A method like "Weight Selection", which initializes a smaller model by directly copying a subset of the larger model's FFN weights, is a crucial point of comparison.

- Incremental Novelty: The paper's claim that this problem is "unexplored" is an overstatement. The paper explicitly adapts the "Learngene" framework from vision and builds on the known concept of FFNs as knowledge-stores. The novelty rests almost entirely on the addition of the $L_{spec}$ loss.

**Questions:**

The "Distillation" baseline  is underspecified. Is this standard response-based distillation (matching output logits) or feature-based distillation? Given MASA operates on intermediate FFN outputs , a comparison to a modern feature-based distillation method  would be a more rigorous and appropriate baseline.

---

> ### Author Response · Authors · 2025-11-20
> **Response to Reviewer #yUAD**
>
> Thank you for your careful reading and insightful questions.
>
> ***Q1***   **Missing Baseline**
>
> ***R1*** Thank you for the suggestions. Following your advice, we added a baseline where the source model’s parameters are directly resized and used to initialize the target model. The results are shown in the table below.
>
> | Baseline | PIQA | medmcQA | Dolly |
> | --- | --- | --- | --- |
> | Scratch | 52.34 | 34.99 | 22.59 |
> | direct parameter transfer | 52.94 | 35.12 | 22.89|
> |MASA | **56.75** | **36.55** |**23.90**|
>
> ***Q2***   **Incremental Novelty: The paper's claim that this problem is "unexplored" is an overstatement.**
>
> ***R2***  Thank you for your question. The proposed approach of extracting **generalizable knowledge** from large models through a **learngene matrix** has indeed not been explored before. Even within the Learngene framework in the computer vision community, prior work typically **directly extracts parameters** from the model as learngenes, without attempting to use an intermediate representation or transformation as we do.
> Therefore, our method, leveraging a mediating learngene matrix to extract and transfer shared knowledge, is genuinely novel and represents an original contribution beyond existing Learngene-based techniques.
>
> ***Q3***   **The "Distillation" baseline is underspecified.**
>
> ***R3***  Thank you for your question. For the distillation baseline used in our comparison, we adopt the **MiniLLM** distillation framework. As for the ​**teacher model**​, we follow the results reported in the **OLMoE** paper. Since **OLMoE-7B** consistently outperforms **DeepSeekMoE-16B** on downstream tasks, we select the stronger **OLMoE-7B** as the teacher model for our distillation baseline.

---

### Author Response · Authors · 2025-11-28
**Request for Guidance on Reviewer-Author Discussion Status**

Dear Area Chair,

I hope you are doing well. I am writing to kindly ask for your guidance regarding the discussion phase of our submission.

At the moment, one reviewer have participated, while the others have not yet responded. We fully understand that reviewers may be busy, and we sincerely appreciate the time and effort they dedicate to the process.

If appropriate, we would be grateful if you could remind the remaining reviewer or let us know whether any further action is needed from our side.

Thank you very much for your time and for your service to the ICLR community.

Warm regards, Authors of Paper 1242

---

### Meta-Review · Area_Chair_S4dj · 2026-01-08

**Summary:**

This paper proposes MASA, a matrix-level framework for transferring generalizable knowledge from LLM FFN layers. By employing a dual-alignment objective—consisting of functional output alignment and structural spectral alignment—MASA enables the scalable initialization of smaller dense and MoE models. Experimental results demonstrate that MASA significantly outperforms random initialization, pruning, and distillation, achieving faster convergence and superior performance with a fraction of the pre-training data.

**Reviewer Concerns:**

1. Reviewer yUAD core concerns were adequately addressed.
+ (methodological concerns) Authors addressed missing baseline (direct parameter transfer) which MASA consistently outperformed.
+ (underspecified distillation baseline concerns) Authors specifies the use of MiniLLM with OLMoE-7B as teacher, removing ambiguity.
+ (incremental novelty concerns): The authors clarified that MASA is not simple parameter copying or pruning, but uses learnable gene matrices with spectral alignment, which is distinct from prior approaches.
2. Reviewer Wu8B remains unconvinced.
+ Novelty skepticism relative to pruning/distillation
+Theoretical depth: spectral alignment motivation is heuristic
3. Reviewer Nd6r wanted zero-shot or few-shot evaluation and the authors argued that this is not meaningful for 570M models- this concerns remains partially unresolved.
4. Reviewer KeEV concerns regarding definition of "highly active experts" is fully resolved and baseline comparability and implementation details is clarified. Terminology and presentation issues are acknowledged and fixed.

The following concerns were addressed by rebuttal:
+ Missing and underspecified baselines
+ Distillation teacher/model ambiguity
+ Expert selection criteria in MoE models
+ Hyperparameter sensitivity
+ Several clarity and implementation-detail issues

Still outstanding
+Degree of conceptual novelty vs learngene
+Evaluation scope for generalizable knowledge

**Reviewer Scores:**

yUAD - 4,
Wu8B -2
Nd6r - 4
KeEV -4

Majority of the reviewers would voted borderline accept and one strong negative.

---

### Decision · Program_Chairs · 2026-01-26

Accept (Poster)